# Topotactic fabrication of transition metal dichalcogenide superconducting nanocircuits

Xiaohan Wang[1,5], Hao Wang[1,2,5] ✉, Liang Ma[1,5], Labao Zhang[1,2] ✉, Zhuolin Yang[1], Daxing Dong[3], Xi Chen[4], Haochen Li[1], Yanqiu Guan[1], Biao Zhang[1], Qi Chen[1], Lili Shi[1], Hui Li[1], Zhi Qin[1], Xuecou Tu[1], Lijian Zhang[1], Xiaoqing Jia[1,2], Jian Chen[1], Lin Kang[1,2] & Peiheng Wu[1,2] ✉

Superconducting nanocircuits, which are usually fabricated from superconductor films, are the core of superconducting electronic devices. While emerging transition-metal dichalcogenide superconductors (TMDSCs) with exotic properties show promise for exploiting new superconducting mechanisms and applications, their environmental instability leads to a substantial challenge for the nondestructive preparation of TMDSC nanocircuits. Here, we report a universal strategy to fabricate TMDSC nanopatterns via a topotactic conversion method using prepatterned metals as precursors. Typically, robust $NbSe_2$ meandering nanowires can be controllably manufactured on a wafer scale, by which a superconducting nanowire circuit is principally demonstrated toward potential single photon detection. Moreover, versatile superconducting nanocircuits, *e.g.*, periodical circle/triangle hole arrays and spiral nanowires, can be prepared with selected TMD materials ($NbS_2$, $TiSe_2$, or $MoTe_2$). This work provides a generic approach for fabricating nondestructive TMDSC nanocircuits with precise control, which paves the way for the application of TMDSCs in future electronics.

Since the first discovery of intrinsic superconductivity in exfoliated 2H-$NbSe_2$ in the 1970s[1], 2D transition metal dichalcogenide superconductors (TMDSCs) have drawn considerable research interest and expedited insight into novel physical properties[2–9]. Van der Waals heterostructures based on TMDSC mono/few layers have also spawned exotic superconducting phenomena and new physical mechanisms[10–13]. These novel physical phenomena have also derived many superconducting electronic devices, such as photodetectors[14], nonreciprocal antennas[15], and supercurrent diodes[16]. Nanopatterning fabrication is the critical gap in the development of TMDSCs from fundamental research to practical applications[16–20]. Notably, recent advances in fundamental studies on TMDSCs are predominantly

conducted on mechanically exfoliated flakes and/or restacked heterostructures, which are still far from practical device applications due to their limited size (<100 μm) and suboptimal stability[3,5,21].

Recently, large-area TMDSCs have been intensely pursued through a variety of progressive approaches, including chemical intercalation and exfoliation[22], chemical vapor deposition (CVD)[23,24], and molecular beam epitaxy (MBE)[25,26]. Strikingly, Lin et al.[24] reported a two-step growth of environmentally stable wafer-scale TMDSC films, laying the foundation for the future development of integrated superconducting electronic devices. Superconducting nanocircuits patterned from films play an indispensable role in the function and performance of superconducting electronic devices[14,27–33]. For

[1]Research Institute of Superconductor Electronics, School of Electronic Science and Engineering, College of Engineering and Applied Science, Nanjing University, Nanjing 210023, China. [2]Hefei National Laboratory, Hefei 230088, China. [3]Department of Applied Physics, Nanjing University of Aeronautics and Astronautics, Nanjing 210016, China. [4]Department of Physics, Tsinghua University, Beijing 100084, China. [5]These authors contributed equally: Xiaohan Wang, Hao Wang, Liang Ma. ✉e-mail: wanghao91@nju.edu.cn; Lzhang@nju.edu.cn; phwu@nju.edu.cn

example, in superconducting nanowire single-photon detectors (SNSPDs), ultrathin superconducting films need to be etched into meandered nanowires for ultrahigh detection sensitivity[34,35]. In addition, patterned superconducting systems have induced many exotic physics, e.g., bosonic metallicity[36], and metallic TMDSC gratings[37] have been demonstrated to be promising candidates for the near-infrared range. However, despite inspiring achievements, the nondestructive patterning of superconducting nanocircuits on TMDSC films has not been well demonstrated. The multi-step patterning process, involving electron beam lithography and reactive ion etching, inevitably destroys the environmentally unstable TMDSC films. Thus, to advance practical applications in integrated electronics, the development of nondestructive processing of TMDSC nanocircuits is critical but challenging.

Here, we devise a topotactic conversion approach to achieve the nondestructive fabrication of various TMDSC nanocircuits through the chalcogenization of prepatterned transition metals. Typically, multifarious structures (e.g., meandering/spiral nanowires and periodical holey arrays) are patterned on transition metals by electron beam lithography and reaction ion etching, which topotactically transform into targeted TMDSC nanocircuits after annealing in chalcogen atmospheres. A set of microscopies, spectroscopic, and electrical characterizations reveal that the topotactically fabricated NbSe$_2$ nanowires retain better structural and superconducting properties compared to the counterpart obtained from patterning NbSe$_2$ films. A demoed device on a NbSe$_2$ circuit delivers regular critical currents in different widths under different temperatures, demonstrating its high quality and potential for further application in integrated electronics.

## Results

### Strategies for preparing TMDSC nanocircuits

TMDSCs have many novel phenomena in quantum physics and show potential in both fundamental research and practical applications, for which the fabrication of TMDSC nanocircuits is essential. The traditional top-down patterning strategy that patterns a film into the targeted nanocircuit has been widely applied in the electronic community. However, this process involves a set of chemical treatments, including resist coating, baking, developing, and lift-off in organics, which inevitably destroy the environmentally unstable

TMDSCs. As illustrated in Fig. 1, a TMDSC film is first synthesized by topotactic conversion of the metal film sputtered on a substrate[38]. Then, the as-prepared film is top-down patterned into specific nanocircuits, e.g., meandered nanowires. This process could result in the generation of oxidized species and vacancies in the TMDSC nanocircuit, which largely deteriorate its structural and superconducting properties. To meet this challenge, a nondestructive topotactic fabrication method is developed. In detail, the metal film is first patterned into selected nanostructures and then chalcogenized into a nanostructured TMDSC through a topotactic conversion procedure. Impressively, all the patterning processes are applied forward to the metal precursors rather than TMDSCs; as a result, nondestructive fabrication of TMDSC nanocircuits can be readily realized.

### Characterization of NbSe$_2$ meandered nanowire

As a prototypical TMDSC, 2H-NbSe$_2$ has attracted much attention due to its unique out-of-plane or Ising spin-orbit copling[4]. Here, the nondestructive fabrication of a NbSe$_2$ meandered nanowire is demonstrated by the proposed topotactic fabrication strategy (defined as TF-NbSe$_2$). First, a dense Nb film with a thickness of 2 nm (roughness of 0.2 nm) was deposited on the sapphire substrate by magnetron sputtering (Supplementary Fig. 1a). The 2 nm thick Nb film was metallic, and the resistance decreased as the temperature decreased; however, it cannot exhibit superconducting properties at a temperature of approximately 1.8 K, which should be attributed to surface oxidation (Supplementary Fig. 1b)[39]. The X-ray photoemission spectroscopy (XPS) and time-of-flight secondary ion mass spectrometry results confirm that Nb was partially oxidized into a composite of Nb and Nb$_2$O$_5$ after patterning by electron beam lithography and reaction ion etching; consequently, a partially oxidized Nb meandered nanowire with a width of $200 \pm 10$ nm was prepared (Supplementary Fig. 1c–f). The partially oxidized Nb meandered nanowire was transformed into a NbSe$_2$ meandered nanowire by topotactic selenization. In the Raman spectrum (Supplementary Fig. 2a), two peaks at 229 and 243 cm$^{-1}$ can be assigned to the A$_{1g}$ and E$^1_{2g}$ modes of NbSe$_2$, respectively, indicating the topotactic transformation of the Nb precursor into NbSe$_2$[19,40]. The scanning electron microscopy (SEM) and atomic force microscopy (AFM) images (Fig. 2a and Supplementary Fig. 2b–d) show that a continuous meandered NbSe$_2$ nanowire is obtained, which has a dense surface and a uniform linewidth. Note that the height of the NbSe$_2$

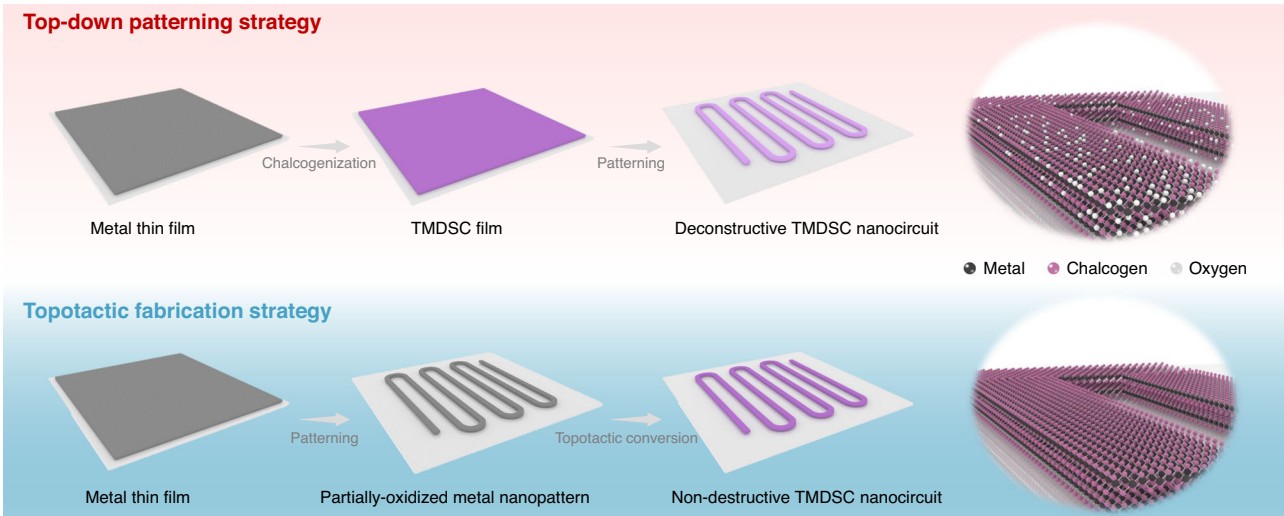

**Fig. 1 | Schematic illustrations of top-down patterning and topotactic fabrication strategies for preparing transition metal dichalcogenide superconductors (TMDSC) nanocircuits.** The top-down patterning method involves three steps: (i) deposition of metal films on a substrate; (ii) chalcogenization of metal films into TMDSC films; and (iii) etching of TMDSC films into nanopatterns with severe degradation. In contrast, the topotactic fabrication method topotactically chalcogenizes the prefabricated partially-oxidized metal nanopatterns, which avoids the destruction of the TMDSCs from the patterning process.

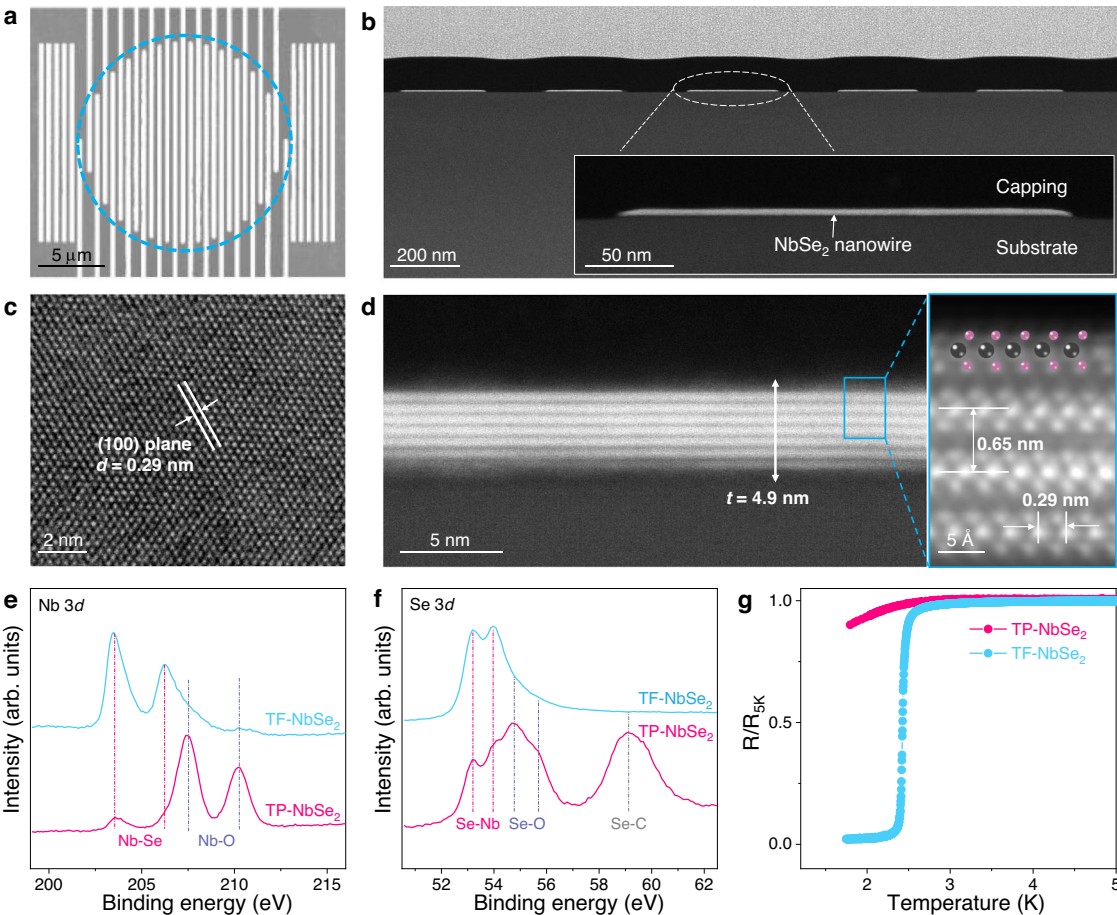

**Fig. 2 | Characterization of NbSe₂ meandered nanowire prepared by topotactic fabrication strategy (TF-NbSe₂). a** Scanning electron microscopy (SEM) image of TF-NbSe₂, in which a gray meandered nanowire on a white background is highlighted by a dashed blue circle. **b** Cross-sectional scanning transmission electron microscopy (STEM) image of TF-NbSe₂ nanowire and inset is the enlarged image of a single TF-NbSe₂ nanowire. **c** In-plane high-resolution transmission electron microscopy (HRTEM) image of TF-NbSe₂ with a lattice distance ($d$) of 0.29 nm for (100) plane. **d** High-resolution STEM image of TF-NbSe₂ along with enlarged atomic-resolution display, showing the van der Waals layered structure with a thickness ($t$) of 4.9 nm and a layer distance of 0.65 nm. The Nb and Se atoms are represented with black and pink spheres, respectively. Core-level Nb 3$d$ (**e**) and Se 3$d$ (**f**) X-ray photoemission spectroscopy (XPS) spectra of TF-NbSe₂ and NbSe₂ prepared by top-down patterning strategy (TP-NbSe₂). The vertical dash lines refer to the bonds below in the same color. **g** Temperature dependence of the resistance for TF-NbSe₂ and TP-NbSe₂. $R_{5K}$ means the resistance at the temperature of 5 K.

nanowire is determined to be ~5 nm, which is 2.2-fold that of the partially-oxidized Nb precursor (2.3 nm). The wafer photography and XRD results demonstrate that the prepatterned Nb precursor is fully and macroscopically transformed into NbSe₂ (Supplementary Fig. 3). Furthermore, the thickness and width of the NbSe₂ nanowires can be precisely controlled by tuning the sputtering time and electron beam lithography parameters (Supplementary Fig. 4).

Aberration-corrected scanning transmission electron microscopy (STEM) was used to reveal the microstructure of NbSe₂. As shown in Fig. 2b, the cross-sectional image of the meandered NbSe₂ nanowire shows a uniform width of 200 nm, and the enlarged image shows the continuity and homogeneity of the TF-NbSe₂. The in-plane high-resolution transmission electron microscopy (HRTEM) image (Fig. 2c) reveals the in-plane atomic arrangement of NbSe₂, in which the lattice distances of 0.29 nm corresponds to the (100) plane. The high-resolution STEM image of TF-NbSe₂ (Fig. 2d) shows the layered structure and typical atomic arrangement of the NbSe₂ crystal without any oxidized species. The layer-by-layer van der Waals structure of NbSe₂, with a layer thickness of 0.65 nm and a total thickness of 4.9 nm[25,41], is in agreement with the AFM result. Even at the edge of the TF-NbSe₂ nanowire, the layered structure is clearly observed without distinguishable oxidation (Supplementary Fig. 5). The corresponding EDS mappings further confirm the intact

NbSe₂ composition at the edge of TF-NbSe₂ nanowire with a negligible oxygen signal. For comparison, another meandered NbSe₂ nanowire is prepared by etching a NbSe₂ film, which is derived from traditional top-down patterning (defined as TP-NbSe₂) (Supplementary Fig. 6).

While the two kinds of meandered NbSe₂ nanowires exhibit similar morphologies, their intrinsic properties vary extremely. First, the chemical state and composition of the two samples are examined by XPS. As shown in Fig. 2e, two pairs of peaks at 203.4/206.2 and 207.5/210.3 eV are observed in the Nb 3$d$ XPS spectrum of the TF-NbSe₂ sample, which is attributed to Nb⁴⁺ (NbSe₂) and Nb⁵⁺ (Nb₂O₅), respectively[42]. It should be noted that the slight oxidation in TF-NbSe₂ arises from surface oxidation during XPS measurement. In contrast, the peaks referring to Nb₂O₅ species are in the majority of the Nb 3$d$ XPS spectrum for the TP-NbSe₂ sample. This suggests that TP-NbSe₂ experiences severe oxidation during the patterning process. Based on the area of the signal peak after deconvolution (Supplementary Fig. 7), TF-NbSe₂ is determined to be composed of 93.67% NbSe₂ and 6.33% Nb₂O₅, while TP-NbSe₂ contains 26.7% NbSe₂ and 73.3% Nb₂O₅. The Nb/Se/O ratios are calculated to be 1/1.76/0.30 and 1/0.31/2.12 for TF-NbSe₂ and TP-NbSe₂, respectively (Supplementary Table 1). In addition, two pairs of peaks at 53.3/54.0 and 54.6/55.3 eV, corresponding to Se²⁻ (NbSe₂) and Se⁴⁺ (SeO₂), respectively, are present in the Se 3$d$ XPS

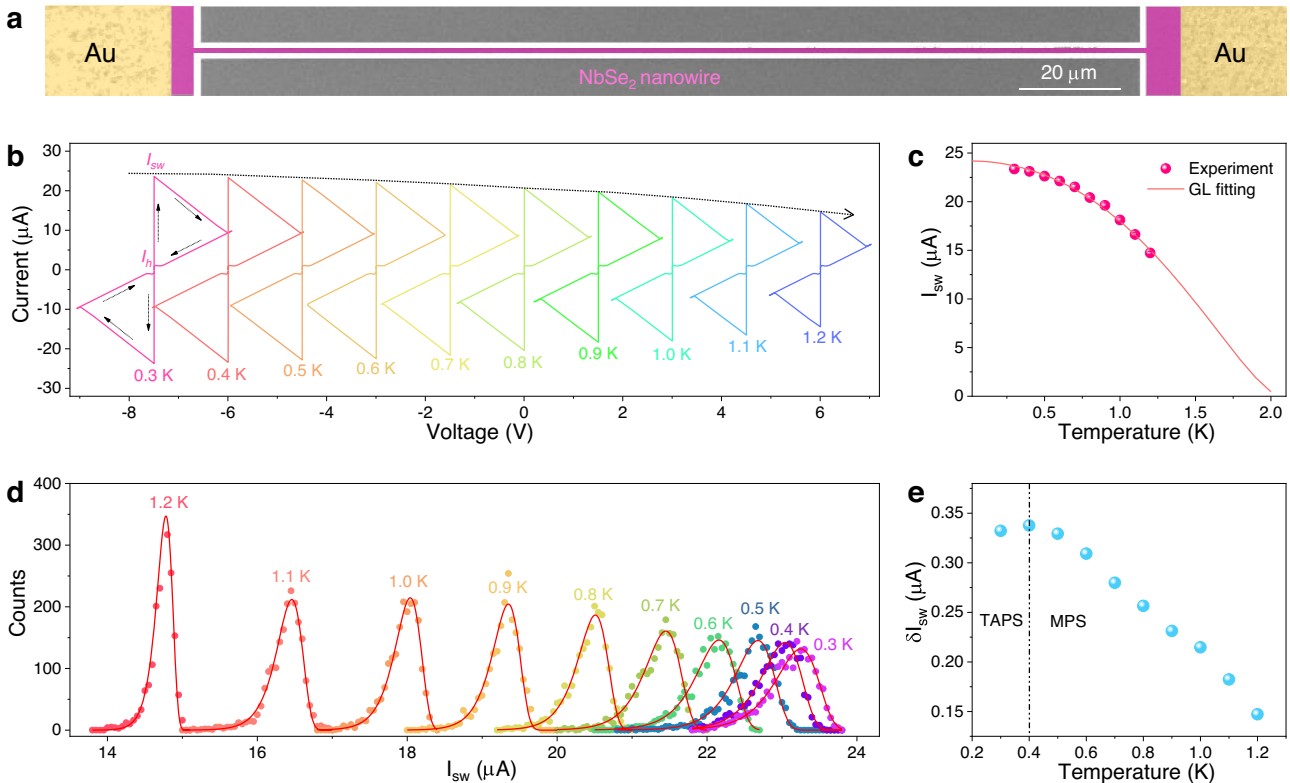

**Fig. 3 | Superconducting properties of TF-NbSe$_2$ nanowire device. a** SEM image of a 200-nm-wide NbSe$_2$ nanowire. **b** $I$–$V$ curves of 200-nm-wide NbSe$_2$ device under different temperatures. The curves are horizontally offset for clarity, and the arrows represent the bias current ($I_b$) scan order. **c** The average values of switch current ($I_{sw}$) for 200-nm-wide NbSe$_2$ device under different temperatures, which can be fitted by the Ginzburg–Landau model. **d** Distribution of $I_{sw}$ values for 200-nm-wide NbSe$_2$ device at different temperatures, which follows Gumbel fitting. **e** Standard deviation $I_{sw}$ ($\delta I_{sw}$) of 200-nm-wide NbSe$_2$ device under different temperatures, in which 400 mK is the crossover temperature of thermally active phased-slip (TAPS) and multiple phase-slip (MPS).

spectra of both TF-NbSe$_2$ and TP-NbSe$_2$ (Fig. 2f)[43]. Obviously, the proportion of SeO$_2$ in TP-NbSe$_2$ is much higher than that in TF-NbSe$_2$, indicating even more serious oxidation of NbSe$_2$, which is consistent with Nb 3$d$ XPS analyses. Notably, one more peak at 58.9 eV attributed to the C–Se bond emerges in TP-NbSe$_2$[44]. It is speculated that the organic resist applied in the patterning process can react with NbSe$_2$ and form C–Se species, which will undoubtedly deteriorate the quality of the NbSe$_2$ nanocircuit. Notably, TF-NbSe$_2$ shows competitive quality in structure and composition to NbSe$_2$ film prepared by selenization of Nb film (Supplementary Fig. 8), indicating that partial oxidation of Nb nanopattern hardly affects the TF-NbSe$_2$ quality.

To further characterize the superconducting quality, the transport properties at low temperatures of TP-NbSe$_2$ and TF-NbSe$_2$ are compared. Figure 2g shows the temperature dependence of the normalized longitudinal resistance at zero magnetic field. For TF-NbSe$_2$, its resistance begins to plunge at ~2.7 K and reaches zero at ~2.2 K, suggesting the trigger of superconductivity. In contrast, the resistance of TP-NbSe$_2$ only slightly decreases as the temperature decreases from 2.9 to 1.8 K, corresponding to a nonsuperconducting state. A TP-NbSe$_2$ nanowire reported by Mills et al. exhibited a 31.4% decrease in $T_c$ compared to the initial NbSe$_2$ flake, indicating a substantial degradation in the superconductivity caused by the destructive patterning process[19]. Obviously, the NbSe$_2$ nanopatterns obtained by the topotactic fabrication strategy can retain excellent superconductivity compared to that from top-down patterning.

### Demonstration of TF-NbSe$_2$ nanowire devices

To investigate the feasibility of the topotactic fabrication strategy, a superconducting device based on TF-NbSe$_2$ nanowires is demoed.

Figure 3a displays the SEM image of the 200-nm-wide NbSe$_2$ nanowire device with one electrode side connected to the ground electrode (GND), which is systematically evaluated by current–voltage ($I$–$V$) characteristics at low temperatures. A measuring circuit is needed to conduct the electrical measurements. As shown in Supplementary Fig. 9, the adjustable voltage source meter ($V_S$) series connection with resistor R$_O$ (100 kΩ) provides variable currents to the devices; thus, the bias current ($I_b$) is input to the DC terminal of the bias-tee and then to the device[45]. Figure 3b displays the $I$–$V$ curves of the 200-nm-wide NbSe$_2$ device at different temperatures with a current sweep rate of 0.2 μA s$^{-1}$. Clearly, the device remains in the superconducting state until the $I_b$ exceeds the switch current ($I_{sw}$, where the superconducting device transitions to a nonsuperconducting state). Once $I_b$>$I_{sw}$, the device immediately transforms to a normal state and generates resistance. With decreasing $I_b$ below $I_{sw}$, the device begins with a normal state due to residual Joule heating and finally recovers to the superconducting state at the hysteresis current ($I_h$). Among them, the $I_{sw}$ value, relative to the number of Cooper pairs, increases with decreasing operation temperature, and the $I_h$, immune to noise, reflects the equilibrium state between Joule heat and heat dissipation in the nanowire[46]. The $I_{sw}$/$I_h$ is determined to be over 21 at 1 K (Supplementary Table 2), which is higher than those reported for superconducting nanowires[47], suggesting promise for application in superconducting single-photon detectors. The relationship between $I_{sw}$ mean values and operation temperature is plotted in Fig. 3c, which can be fitted according to the Ginzburg–Landau (GL) theory[48]. The good agreement between the experimental data and the fittings suggests that the superconductivity of the 200-nm-wide NbSe$_2$ nanowire basically obeys GL theory, and the $T_c$ of the 200-

nm-wide $NbSe_2$ nanowire is determined to be approximately 2 K by fitting the GL theory.

Furthermore, phase fluctuations reveal the intrinsic fluctuation of the superconductors and are the cause of the superconducting transition, representing the stability of the system under certain temperatures, that is, the distribution of $I_{sw}$ when the superconductor switches to the normal state. Here, we further performed 2000 $I-V$ sweeps with a sweep rate of $5\,nA\,s^{-1}$ to record the distribution of $I_{sw}$ values of the 200-nm-wide $NbSe_2$ device at different operation temperatures (Supplementary Fig. 10). After frequency counting with a bin size of 50 nA, all the $I_{sw}$ distributions are basically consistent with the Gumbel fitting from 300 mK to 1.2 K (Fig. 3d)[49]. The standard deviation $\delta I_{sw}$ of each distribution under a certain temperature is calculated and plotted in Fig. 3e. The overall trend follows the conventional phase-slip theory, and $\delta I_{sw}$ reaches a peak at 400 mK, which decreases with the operating temperature from 400 mK to 1.2 K and slightly decreases after the temperature is below 400 mK. This indicates that 400 mK is the crossover temperature of the thermally active phase-slip (TAPS) and multiple phase-slip (MPS) for the $NbSe_2$ device[46,50]. Moreover, $NbSe_2$ nanowire devices with different widths of 200, 400, 800, and 1000 nm are fabricated and compared (Supplementary Fig. 11). All the relationships between $I_{sw}$ and operation temperature for the other three devices are in accordance with GL theory (Supplementary Fig. 12).

Based on the abovementioned analyses, the traditional top-down patterning approach cannot meet the nondestructive fabrication of TMDSC nanocircuits, which usually suffer from heavy degradation during chemical treatments. To this end, a topotactic fabrication strategy is developed, by which nondestructive TMD (with $NbSe_2$ as a typical instance) superconducting nanocircuits with tunable parameters (e.g., thickness and width) can be produced through topotactical chalcogenization of prepatterned metal precursors. Comparative characterizations demonstrate the superior structural and superconducting properties of the TF-$NbSe_2$ nanocircuit. Moreover, a superconducting device is demoed by TF-$NbSe_2$, exhibiting a typical BCS superconductor. It can be concluded that the topotactic fabrication strategy proposed in this work can overcome the bottleneck of the traditional top-down patterning method when facing the fabrication of environmentally unstable TMDSCs.

## The universality of topotactic fabrication strategy

The topotactic fabrication strategy can prevent TMDSCs from suffering damage during the patterning process and realize the nondestructive fabrication of TMDSC nanocircuits for potential device applications. The meandered $NbSe_2$ superconducting nanowire reaches a length of over 200 μm with superior continuity and homogeneity. By this topotactic fabrication strategy, the length of nanocircuits is determined by the prepatterning process but is independent of the TMD materials. Strikingly, this method is widely applicable to the nondestructive preparation of diverse TMDSC nanocircuits in different structures and compositions. For structure, holey patterning of superconductors provides a platform to exploit exotic superconductivity fundamentally, e.g., bosonic metallicity[36]. For material, $TiSe_2$ is another representative TMDSC with inspiring superconductivity and charge density waves[51,52]. To prove the universality of the topotactic fabrication strategy, superconducting nanocircuits of circle-holey $NbS_2$, triangle-holey $TiSe_2$, and spiral $MoTe_2$ nanowires are fabricated (Fig. 4 and Supplementary Fig. 13). It is undoubted that this strategy is also applicable to TMD semiconductor nanopatterns for integrated devices (Supplementary Fig. 14). These instances support the promise of the topotactic fabrication strategy in manufacturing high-quality TMDSC nanocircuits for both fundamental research and practical applications.

## Discussion

In summary, we have demonstrated a facile and universal strategy for approaching the nondestructive fabrication of TMDSC nanocircuits, including meandered/spiral nanowires and circle/triangle-holey arrays, based on a topotactic conversion mechanism. The strategy employs topotactic chalcogenization of prepatterned metal precursors, which can effectively avoid damage from the patterning process and retain the superconductivity of the TMDSC nanocircuits (demoed by $NbSe_2$ nanowire devices). Notably, this approach enables the preparation of diverse TMDSC nanocircuits with tunable thickness and width. As such, this scalable fabrication technique ensures the nondestructive production of high-quality TMDSC nanocircuits with precise control. It is beyond doubt that with further development and optimization, more kinds of structures (e.g., in-plane/out-of-plane heterostructures) and compositions (e.g., ternary/quaternary alloys) with striking superconducting properties could be accessible, which will promote the practical application of TMDSCs in functional electronics.

## Methods

### Fabrication of $NbSe_2$ meandered nanowires

A topotactic fabrication strategy was developed to prepare nondestructive $NbSe_2$ meandered nanowires. First, Nb thin films were deposited on single-side polished $c$-plane sapphire substrates (Hefei Kejing) by magnetron sputtering (base pressure $<9 \times 10^{-6}$ Pa). During the sputtering process, the working power, working pressure, gas flow rate, and deposition rate were set to 100 W, 3 mTorr, 80 sccm, and $0.2\,nm\,s^{-1}$, respectively.

Second, the sputtered Nb films were patterned into meandered nanowires. A layer of polymethyl methacrylate (PMMA, Allresist) electron beam resist was spin-coated on the Nb film. After exposure by E-beam lithography (EBL, Raith), the film was developed in methyl isobutyl ketone (MIBK, Allresist) for 90 s and isopropanol (Aladdin, 99%) for 60 s to obtain the desired meandered nanowire structure on PMMA. Then, the exposed film was etched for 40 s by reactive ion etching (RIE, Samco RIE-10NR) with a $CF_4$ flow of 40 sccm, a working pressure of 4 Pa, and a working power of 100 W. The residual PMMA was removed through a water bath with N-methyl pyrrolidone (Aladdin, 99%) at 80 °C for 30 min.

Finally, a two-zone furnace was used for the topotactic selenization of the patterned Nb nanowires. Selenium powder (500 mg) and patterned Nb sample were placed at zones I (upstream) and II (downstream), which were heated to 450 and 800 °C, respectively. Ar (50 sccm) and $H_2$ (50 sccm) were used as carrier gases. After annealing for 6 min, the furnace was rapidly cooled to room temperature.

For comparison, $NbSe_2$ meandered nanowires were also prepared by a top-down patterning method. The as-sputtered Nb films were first selenized into $NbSe_2$ films, followed by patterning into meandered nanowires.

The topotactic fabrication strategy can be adapted to generally fabricate various TMDSC nanopatterns, including circle-holey $NbS_2$, triangle-holey $TiSe_2$, and spiral $MoTe_2$ nanowires. The detailed parameters are listed in Supplementary Tables 3 and 4.

### Characterizations

SEM and AFM images were captured by Compact Melin (Zeiss) and Dimension Icon (Bruker), respectively. TEM images were captured by an FEI TECNAI G2 F20 200 kV. Cross-sectional HAADF-STEM images were captured by a Titan Cubed G2 60−300 system. XPS spectra were collected with a Thermo Fisher Es-calab. Raman spectra were obtained on a LabRAM HR Evolution (Horiba) with a 532 nm laser. All samples were fabricated by the EBL system (Raith, EBPG5200) and RIE (Samco RIE-10NR).

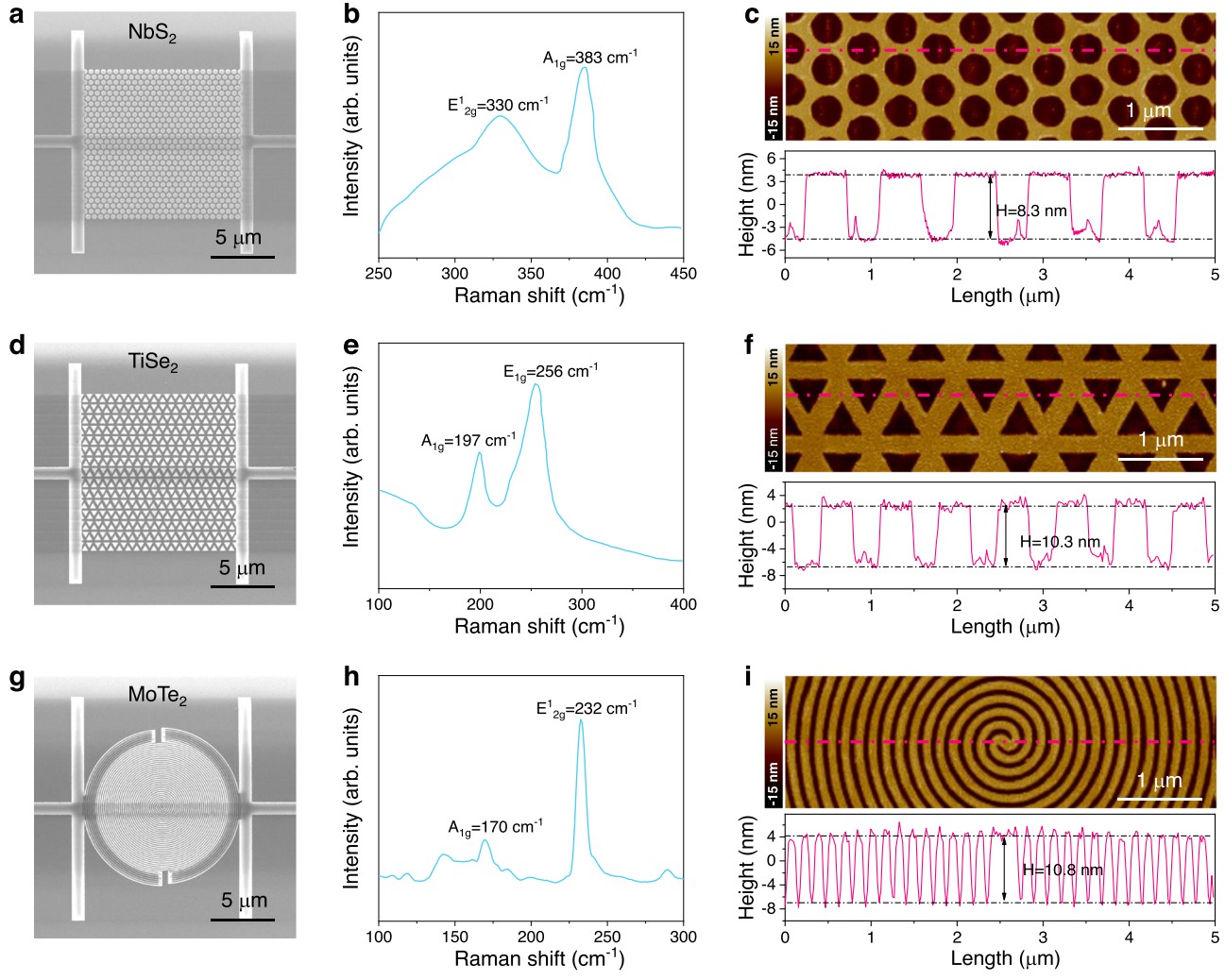

**Fig. 4 | Universality of topotactic fabrication strategy.** SEM images, Raman spectra, and AFM images of circle-holey $NbS_2$ (**a**–**c**), triangle-holey $TiSe_2$ (**d**–**f**), and spiral $MoTe_2$ nanowires (**g**–**i**). The height ($H$) profiles in **c**, **f**, and **i** are derived from the dashed red line in relative AFM images.

## Electrical measurements

The electrical transport characteristics of devices were carried out in the Dilution refrigerating machine for Trition 9 (Oxford Instruments). The measuring circuit includes a digital source meter (keysight 2901) and a T-shaped bias tee. The RF&DC port of the T-shaped Bias-tee is connected to the coaxial cable, leading to devices under extremely low temperatures. The DC port is connected to a digital source meter and the resistor $R_0$ (100 kΩ) to provide a constant current source, and the RF port is connected to an $R_1$ (50 Ω) load resistance. Before the measurements, spot welding is used to connect the Ti/Au electrodes to the coaxial cable. The detailed method and corresponding real pictures are presented in Supplementary Fig. 15.

## Data availability

The Source Data underlying the figures of this study are provided in the paper. All raw data generated during the current study are available from the corresponding authors upon request. Source data are provided in this paper.

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

## Acknowledgements

This work was supported by the Natural Science Foundation of Jiangsu Province (No. BK20210177 (H.W.)), the Innovation Program for Quantum Science and Technology (No. 2021ZD0303401 (L.K.)), the National Natural Science Foundation of China (Nos. 12033002 (L.Z.), 62101240 (H.W.), 62071218 (X.J.), and 62288101 (P.W.)), the Key-Area Research and Development Program of Guangdong Province (2020B0303020001 (L.Z.)), the Priority Academic Program Development of Jiangsu Higher Education Institutions (PAPD) and the Jiangsu Provincial Key Laboratory of Advanced Manipulating Technique of Electromagnetic Waves.

## Author contributions

H.W., La.Z., and P.W. conceived the study. X.W., L.M., Z.Y., Ha.L., Y.G., B.Z., Q.C., L.S., Hu.L., and Z.Q. performed experiments. D.D., X.C., X.T., Li.Z., X.J., J.C., and L.K. analyzed data. X.W., H.W., La.Z., and P.W. wrote the paper, with all authors editing and approving the final version.

## Competing interests

The authors declare no competing interests.
