## [Peer Review File · Nature Communications]

Topotactic fabrication of transition metal dichalcogenide superconducting nanocircuitsREVIEWER COMMENTS

Reviewer #1 (Remarks to the Author):

Manuscript ID: NCOMMS-23-10945-T

Manuscript Title: Topotactic fabrication of transition metal dichalcogenide superconductor nanocircuits

Transition metal dichalcogenides (TMD) materials have a rich crystal structure, leading to plentiful physical properties. Superconductor nanocircuits, which are usually fabricated from superconductor films, are the core of 17 superconducting electronic devices. While emerging transition-metal dichalcogenide superconductors 18 (TMDSCs) with exotic properties show promise for exploiting new superconducting mechanisms and 19 applications, their environmental instability leads to a substantial challenge for the nondestructive 20 preparation of TMDSC nanocircuits. Here, we report a universal strategy to fabricate TMDSC 21 nanopatterns via a topotactic transformation method using prepatterned metals as precursors.

These data are helpful for readers who are interested in TMD materials. Therefore, I suggest it can be accepted after fixing the following issues.

- (1) Figure S1 B and Figure 2f. As we know, The bulk Nb material shows a superconducting transition around 7 K. Why there is no superconductivity below 2 K for the 2 nm thick Nb film?
- (2) Usually, how long can nano TMD superconducting materials be made by the reported method in this work? Is the length of these nano TMD superconducting materials dependent?
- (3) The top-down patterning and topotactic fabrication strategies can be used for all the TMD materials?
- (4) Can the authors give a detailed method and real pictures for measuring the superconducting properties, including the resistivity and magnetic susceptibility?
- (5) Some related literature should be not ignored in the introduction, such as J. Phys.: Condens. Matt., 2019, 32(2), 025702, Nat Phys. 2022, 18, 1425, Phys. Rev. Lett. 2016, 117, 106801, and so on.

Reviewer #2 (Remarks to the Author):

This manuscript presented an efficient bottom-up approach to fabricating TMDSC nanocircuits by chalcogenizing prepatterned transition metal circuits. The authors have provided well-written and well-organized content, and their main argument is backed by sufficient data. Despite these strengths, there are some major issues that need to be addressed before this paper can be published in Nat. Commun., as shown as followings:

1. The first concern pertains to the potential degradation of the transition metal during the prepattern process, which includes EBL, baking, and lift-off in organics. In order to address this concern, comparative experimental data and a theoretical basis should be provided to eliminate the possibility of metal degradation.
2. It is suggested to provide additional evidence, on a macroscopic level, to demonstrate that Nb has been transformed into NbSe₂, rather than forming Nb/NbSe₂ heterostructures. While cross-sectional HRTEM offers microscopic evidence, a Raman spectrum provides a macroscopic view but may not rule out the possibility of forming heterostructures.
3. It is important to establish the relationship between chalcogenization parameters and the thickness of the prepatterned transition metal, which would indicate the precise control characteristics of this strategy.
4. In order to enhance the manuscript, it is recommended to provide a comparison of the superconducting properties between the TF-NbSe₂ developed in this study and TP-NbSe₂ reported in previous literature.

Reviewer #3 (Remarks to the Author):

Superconductor nanocircuits based on transition metal dichalcogenide are promising for next-generation superconducting applications but lack environmental stability. In this manuscript, the authors developed a nondestructive topotactic fabrication approach for NbSe₂ superconductor nanocircuits, compared it with the traditional top-down method, and used it for other TMD nanocircuits as well.

Although the study looks very interesting and can be applied directly to other research, there are already many reports showing the growth of environmentally stable transition metal selenide films, such as

Nat. Mater. 18, 602–607 (2019), where NbSe₂ films can withstand high temperatures and different types of treatment but still preserve the superconducting behaviours. Therefore, the author's method would be less meaningful for high-quality TMD materials. In addition, the chalcogenization method of pre-patterned transition metal is well-known for TMD material growth. Although changing the order of fabrication (patterning before growth) can improve the quality of the final superconductor nanocircuit, this methodology is not substantially advanced over previous works.

Moreover, there are still some concerns I would like the author to address:

1. The author demonstrated the process of top-down patterning (TP) and topotactic fabrication (TF) methods in Figure 1. The author claims the patterning process will induce oxidized species and vacancies in the sample. However, the chemical treatments during the patterning process can also induce oxidization of the Nb metal, forming amorphous NbO_x before the chalcogenization process. This pre-contamination of the Nb precursor could further impact the quality of the final NbSe₂ materials. The author needs to clarify the impact of patterning on the Nb precursor and the final NbSe₂ products.

2. In Figure 2b-c, the author demonstrated HRTEM images of multilayer TF-NbSe₂. How homogeneous are these results? Does the author also find any region with oxidization? How about the edges of the nanowires (most likely to be influenced by the patterning process)?

3. In Figure 2d-e, the author compared the XPS results of TF-NbSe₂ and TP-NbSe₂ and noticed substantial oxidization in the TP-NbSe₂. Can the author also perform a concentration analysis and compare the Nb, Se and O ratios?

4. In Figure 2f, the author demonstrated superconducting behaviors of TF-NbSe₂ and non-superconducting behaviors of TP-NbSe₂. Since the patterning process are almost inevitable during the circuit fabrication process, how do these TP-NbSe₂ results compare to other reports about superconductor nanocircuits patterned from films? Is it possible that other film growth methods (CVD, MOCVD) produce better-quality NbSe₂ materials and, therefore, still achieve good superconductivity behaviour after patterning?

5. In most superconductivity measurements, voltage is plotted against current (such as Science, 372(6540), pp.409-412.), and at a certain current, we see a sharp increase in voltage. Higher than that, the circuit shows as a resistor. In this manuscript, Figure 3b plots the current against voltage and demonstrates a sharp increase in current at a certain voltage. Is there a particular reason for that?

Point-by-point response to the reviewers' comments

Reviewer #1

General Comment: Transition metal dichalcogenides (TMD) materials have a rich crystal structure, leading to plentiful physical properties. Superconductor nanocircuits, which are usually fabricated from superconductor films, are the core of superconducting electronic devices. While emerging transition-metal dichalcogenide superconductors (TMDSCs) with exotic properties show promise for exploiting new superconducting mechanisms and applications, their environmental instability leads to a substantial challenge for the nondestructive preparation of TMDSC nanocircuits. Here, we report a universal strategy to fabricate TMDSC nanopatterns via a topotactic transformation method using prepatterned metals as precursors. These data are helpful for readers who are interested in TMD materials. Therefore, I suggest it can be accepted after fixing the following issues.

Response: We thank the reviewer for positive feedback and insightful comments to further improve the quality of our manuscript. We have addressed these comments point by point in the following paragraphs.

Comment 1-1: Figure S1 B and Figure 2f. As we know, the bulk Nb material shows a superconducting transition around 7 K. Why there is no superconductivity below 2 K for the 2 nm thick Nb film?

Response: We agree with the reviewer's comment. For bulk Nb materials, the theoretical superconducting transition temperature (T_c) is up to 9.7 K (*Phys. Rev. B* 2005, 72, 064503). The bulk Nb film (thickness: ~300 nm) on sapphire grown by magnetron sputtering exhibits a T_c of 9.11 K (**Figure R1a**), indicating the high quality of the as-synthesized sample. Due to surface oxidation, which can be demonstrated by XPS analysis (**Figure R1b**), the ultrathin Nb film (thickness < 5 nm) usually shows semimetal behavior rather than superconductivity (*Molecules* 2023, 28, 1059). Thus, we deem that the surface oxidation results in the nonsuperconductivity of the 2 nm thick Nb film below 2 K.

Figure R1. (a) R - T curve of the bulk Nb material. (b) XPS spectrum of the Nb 3d of the 2 nm thick Nb film.

We have added relative discussion in the revised manuscript: “The 2 nm thick Nb film was metallic, and the resistance decreased as the temperature decreased; however, it cannot exhibit superconducting properties at a temperature of

approximately 1.8 K, which should be attributed to surface oxidation (**Supplementary Fig. S1b**)³⁹.” (Line 1, Page 5)

Comment 1-2: Usually, how long can nano TMD superconducting materials be made by the reported method in this work? Is the length of these nano TMD superconducting materials dependent?

Response: We appreciate the reviewer’s constructive comment. The length of the TMD superconducting nanowires can be hundreds of microns in a straight line and millimeters for a meandered line, which is determined by the E-beam lithography and reactive ion etching process. The length of the nanowire is independent of the TMD materials. The meandered NbSe₂ superconducting nanowire (**Figure 2a** in the **Manuscript**) with a length of more than 200 μm exhibits good continuity, homogeneity and superconductivity, which is adequate for developing superconductor devices (e.g., SNSPD).

We have added more discussion in the revised manuscript: “The meandered NbSe₂ superconducting nanowire reaches a length over 200 μm with superior continuity and homogeneity. By this topotactic fabrication strategy, the length of nanocircuits is determined by the pre patterning process but is independent of the TMD materials.” (Line 27, Page 10)

Comment 1-3: The top-down patterning and topotactic fabrication strategies can be used for all the TMD materials?

Response: We thank the reviewer for the profound comment. The top-down patterning strategy is a conventional approach for fabricating various nanocircuits for electronic devices, which is applicable to all the as-grown TMD films. However, the etching process usually destructs the TMD by creating defects and/or oxidized species, which consequently degrades the device performance.

The topotactic fabrication strategy that prepares TMD nanocircuits by the chalcogenization of pre patterned metal films can effectively avoid the destruction of TMD materials from the etching process. In the manuscript, we have demoed four types of TMD superconductor nanocircuits by topotactic fabrication, including meandered NbSe₂ nanowires, circle-hole NbS₂, triangle-hole TiSe₂, and spiral MoTe₂ nanowires (**Figure 4** in the **Manuscript**). Furthermore, we have applied this strategy to TMD semiconductors, such as MoS₂ square arrays and WSe₂ hexagon arrays (**Figure R2**). It is safe to claim that the topotactic fabrication strategy can be used for all TMD materials.

We have added more discussion on the universality of the topotactic fabrication strategy in the revised manuscript: “It is undoubted that this strategy is also applicable to TMD semiconductor nanopatterns for integrated devices (**Supplementary Fig. S14**)”.” (Line 9, Page 11)

Figure R2 (Supplementary Fig. S14). Raman spectra and AFM images of MoS₂ square arrays (a-b) and WSe₂ hexagon arrays (c-d).

Comment 1-4: Can the authors give a detailed method and real pictures for measuring the superconducting properties, including the resistivity and magnetic susceptibility?

Response: According to the reviewer's suggestion, we have added the detailed method and real pictures for the device measurement in the revised manuscript: "The detailed method and corresponding real pictures are presented in **Supplementary Fig. S15.**" (Line 2, Page 14)

Figure R3 (Supplementary Fig. S15). Real pictures for the device measurements. (a) PCB-fixed NbSe₂ device connected to an external circuit by a low-temperature coaxial cable; (b) enlarged image showing wire bonding; (c) *R-T* and (d) *I-V* tests.

The device is set in the refrigerator and connected to an external circuit via a low-temperature coaxial cable, as displayed in **Supplementary Fig. S15a**. The TF-NbSe₂ device is fixed on the sample holder and connected to the electrodes on the printed circuit board by wire bonding, as shown in **Supplementary Fig. S15b**. For the *R-T* measurement, Python is used to record the resistance on the source meter (the constant current is set as 1 μ A) and the corresponding temperature values (**Supplementary Fig. S15c**). For the *I-V* test, an isolated voltage source in series with a bias resistor ($R_0=100$ k Ω) was included, which was attached to the DC port of a Bias-tee. The RF port was terminated with a 50 Ω load resistance (R_1), and the DC&RF port was attached to the cryostat feedthrough attached to coax that leads down to the measured device (**Supplementary Fig. S15d**). The circuit diagram is also schemed in **Supplementary Fig. S9**. The LabView program is used to collect the testing data.

Comment 1-5: Some related literature should be not ignored in the introduction, such as *J. Phys.: Condens. Matt.*, 2019, 32(2), 025702, *Nat Phys.* 2022, 18, 1425, *Phys. Rev. Lett.* 2016, 117, 106801, and so on.

Response: We appreciate the reviewer's helpful suggestion. 1T polytype layered NbSeTe enlarged the family of TMDSCs (*J. Phys.: Condens. Matt.* 2019, 32, 025702). Zhang et al. tailored Ising superconductivity in intercalated bulk NbSe₂ by intercalating bulk crystals with cations from ionic liquids (*Nat. Phys.* 2022, 18, 1425). Xi et al. demonstrated reversible tuning of superconductivity and CDW order in NbSe₂ by an ionic liquid gate (*Phys. Rev. Lett.* 2016, 117, 106801). Considering their significance in the material and physics research of TMDSCs, we have cited these references following the first sentence "Since the first discovery of intrinsic superconductivity in exfoliated 2H-NbSe₂ in the 1970s, 2D transition metal dichalcogenide superconductors (TMDSCs) have drawn considerable research interest and expedited insight into novel physical properties." in the Introduction. (Line 8, Page 2)

Reviewer #2

General Comment: This manuscript presented an efficient bottom-up approach to fabricating TMDSC nanocircuits by chalcogenizing prepatterned transition metal circuits. The authors have provided well-written and well-organized content, and their main argument is backed by sufficient data. Despite these strengths, there are some major issues that need to be addressed before this paper can be published in Nat. Commun., as shown as followings.

Response: We thank the reviewer for positive feedback and insightful comments to further improve the quality of our manuscript. We have addressed these comments point by point in the following paragraphs.

Comment 2-1: The first concern pertains to the potential degradation of the transition metal during the prepattern process, which includes EBL, baking, and lift-off in organics. In order to address this concern, comparative experimental data and a theoretical basis should be provided to eliminate the possibility of metal degradation.

Response: We fully agree with the reviewer that the transition metals are inevitably oxidized during the prepattern process. In the revised manuscript, we have taken this into consideration. First, XPS analysis of the fresh Nb film and the prepatterned Nb sample is conducted. As shown in **Figure R4a**, the Nb 3d XPS spectra of the two samples exhibit two pairs of peaks referring to Nb and Nb₂O₅. The emergence of Nb₂O₅ in the Nb film can be attributed to surface oxidation during the test. The much stronger signals of Nb₂O₅ than that of Nb in prepatterned Nb indicate heavy degradation after the patterning process.

To unveil the surface and vertical chemical composition of the prepatterned Nb, time-of-flight secondary ion mass spectrometry (TOF-SIMS) was performed. **Figure R4b** shows the normalized intensity curves of elements during TOF-SIMS depth profiling. The different fluences of reaching maxima indicate a layered structure of the sample, which is composed of Nb₂O₅, Nb and Al₂O₃ (substrate). Based on the above analyses, it is concluded that the prepatterned Nb is a composite of Nb and Nb₂O₅.

Figure R4 (Supplementary Fig. S1e,f). (a) Core-level Nb 3d spectra of the Nb film before and after patterning. (b) Normalized TOF-SIMS depth profile of the ultrathin Nb film after nanopatterning.

For an accurate description, we have revised the definition of a prepatterned metal as a partially-oxidized metal nanopattern. It is noted that, as shown in **Figure R5**, TF-

NbSe₂ shows competitive quality in structure and composition to NbSe₂ film prepared by selenization of Nb film. It can be concluded that the partially-oxidized Nb nanopattern can be well and thoroughly transformed into a high-quality NbSe₂ nanopattern through topotactic selenization.

Figure R5 (Supplementary Fig. S8). (a) XRD patterns, and (b) Nb 3d and (c) Se 3d XPS spectra of TF-NbSe₂ and NbSe₂ film.

We have added more discussion on the metal degradation and its impact on the NbSe₂ quality in the revised manuscript: “The X-ray photoemission spectroscopy (XPS) and time-of-flight secondary ion mass spectrometry results confirm that Nb was partially oxidized into a composite of Nb and Nb₂O₅ after patterning by electron beam lithography and reaction ion etching; consequently, a partially-oxidized Nb meandered nanowire with a width of 200±10 nm was prepared (Supplementary Fig. S1c-f).” (Line 4, Page 5); “Notably, TF-NbSe₂ shows competitive quality in structure and composition to NbSe₂ film prepared by selenization of Nb film (Supplementary Fig. S8), indicating that partial oxidation of Nb nanopattern hardly affects the TF-NbSe₂ quality.” (Line 23, Page 7)

Comment 2-2: It is suggested to provide additional evidence, on a macroscopic level, to demonstrate that Nb has been transformed into NbSe₂, rather than forming Nb/NbSe₂ heterostructures. While cross-sectional HRTEM offers microscopic evidence, a Raman spectrum provides a macroscopic view but may not rule out the possibility of forming heterostructures.

Response: We thank the reviewer for the helpful suggestion. To rule out the possibility of forming Nb/NbSe₂ heterostructures, we have provided macroscopic photography of TF-NbSe₂ (Figure R6a). Compared to the gray color of the Nb film, TF-NbSe₂ presents a macroscopically homogeneous and transparent red color. Furthermore, as shown in Figure R6b, the XRD pattern of TF-NbSe₂ exhibits the typical (002), (004), (006) and (008) characteristic peaks of NbSe₂ but no signals of Nb. Both the photography and XRD results demonstrate that the prepatterned Nb precursor is fully and macroscopically transformed into NbSe₂ rather than the Nb/NbSe₂ heterostructure.

We have added more discussion on the macroscopic characterization of TF-NbSe₂ in the revised manuscript: “The wafer photography and XRD results demonstrate that the prepatterned Nb precursor is fully and macroscopically transformed into NbSe₂ (Supplementary Fig. S3).” (Line 17, Page 5)

Figure R6 (Supplementary Fig. S3). (a) Macroscopic photography of TF-NbSe₂ and Nb film; (b) The XRD pattern of TF-NbSe₂.

Comment 2-3: It is important to establish the relationship between chalcogenization parameters and the thickness of the prepatterned transition metal, which would indicate the precise control characteristics of this strategy.

Response: We thank the reviewer's helpful suggestion. To indicate the precise control of the topotactic fabrication strategy, we have added **Supplementary Table S4** into the revised manuscript for the detailed parameters of the chalcogenization process, as displayed in **Table R1**.

Table R1 (Supplementary Table S4). Detailed chalcogenization parameters for synthesizing TMDSCs from prepatterned transition metals with different thicknesses.

TMDSCs	Chalcogen Precursors	Carrier gas	Thickness of prepatterned transition metal	Growth temperature	Growth time
NbSe ₂	Se powder (450°C)	Ar/H ₂ (50/50 sccm)	~4 nm (Nb)	800 °C	10 min
			~3 nm (Nb)	800 °C	8 min
			~2 nm (Nb)	800 °C	6 min
NbS ₂	S powder (220°C)	Ar/H ₂ (50/15 sccm)	~3 nm (Nb)	750 °C	10 min
TiSe ₂	Se powder (450°C)	Ar/H ₂ (50/50 sccm)	~4 nm (Ti)	850 °C	8 min
MoTe ₂	Te powder (500°C)	Ar/H ₂ (15/15 sccm)	~4 nm (Mo)	600 °C	12 min

Comment 2-4: In order to enhance the manuscript, it is recommended to provide a comparison of the superconducting properties between the TF-NbSe₂ developed in this study and TP-NbSe₂ reported in previous literature.

Response: We appreciate the reviewer's constructive suggestion for enhancing the manuscript. To the best of our knowledge, very few papers have studied the superconducting properties TP-NbSe₂, which was prepared by patterning of NbSe₂ flakes mechanically exfoliated from a bulk single crystal (*Appl. Phys. Lett.* 2014, 104, 052604). The as-etched TP-NbSe₂ nanowire with a thickness of 9 nm exhibited a **31.4%** decrease in T_c compared to the initial NbSe₂ flake, indicating a substantial degradation in the superconductivity. Moreover, as discussed in the Introduction, the limited size (up to dozens of micrometers) of exfoliated NbSe₂ is far from satisfying practical integrated devices.

We have added more discussion on the comparison between our TF-NbSe₂ and previously reported TP-NbSe₂ in the revised manuscript: “A TP-NbSe₂ nanowire reported by Mills et al. exhibited a 31.4% decrease in T_c compared to the initial NbSe₂ flake, indicating a substantial degradation in the superconductivity caused by the destructive patterning process¹⁹.” (Line 4, Page 8)

Reviewer 3#

General Comment: Superconductor nanocircuits based on transition metal dichalcogenide are promising for next-generation superconducting applications but lack environmental stability. In this manuscript, the authors developed a nondestructive topotactic fabrication approach for NbSe₂ superconductor nanocircuits, compared it with the traditional top-down method, and used it for other TMD nanocircuits as well.

Although the study looks very interesting and can be applied directly to other research, there are already many reports showing the growth of environmentally stable transition metal selenide films, such as *Nat. Mater.* 18, 602–607 (2019), where NbSe₂ films can withstand high temperatures and different types of treatment but still preserve the superconducting behaviours. Therefore, the author's method would be less meaningful for high-quality TMD materials. In addition, the chalcogenization method of pre-patterned transition metal is well-known for TMD material growth. Although changing the order of fabrication (patterning before growth) can improve the quality of the final superconductor nanocircuit, this methodology is not substantially advanced over previous works. Moreover, there are still some concerns I would like the author to address.

Response: We thank the reviewer for offering valuable and profound comments to further improve the quality of our manuscript. We appreciate the reviewer by pointing out that our work is “*very interesting and can be applied directly to other research*”. Regarding the novelty and significance, to the best of our knowledge, this is indeed the first report of a universal approach for the nondestructive fabrication of TMD superconductor nanocircuits. Superconductor nanocircuits are the key element of superconductor electronic devices, such as hot electron bolometers (HEBs), superconductor nanowire single-photon detectors (SNSPDs) and superconducting quantum bits (qubits), which are widely used in astronomical observation and quantum information. Currently, TMD nanocircuits are mainly produced by nanopatterning TMD films, which inevitably suffers from the etching process, and thus degrades the material quality. As the reviewer mentioned, Lin et al. (*Nat. Mater.* 2019, 18, 602) reported the synthesis of an environmentally stable NbSe₂ film through an oxygen-absent deposition method, which preserved superconductivity after different treatments. In the top-down patterning process, the NbSe₂ film is selectively etched into nanopatterns (e.g., nanowires) through E-beam lithography and reactive ion etching. After such oxygen-involved destructive processing, defects and oxygen species are introduced in the NbSe₂ nanowires, which greatly deteriorates the material properties. This phenomenon has been well demonstrated in this manuscript (TP-NbSe₂). Therefore, even for high-quality TMD materials, it is highly meaningful to develop nondestructive patterning methods.

We agree with the reviewer that chalcogenization of prepatterned metal into TMDs has been reported previously, in which Mo-based TMDs are predominant and the size of TMD patterns is mainly micrometer-level (*Nat. Commun.* 2022, 13, 5597; *ACS Nano* 2021, 15, 410; *J. Mater. Chem. C* 2019, 7, 8599; *Nanoscale* 2018, 10, 1056; *ACS Nano* 2016, 10, 573). In comparison, our work shows substantial advances that demonstrate a universal nondestructive patterning strategy to deliver a variety of TMDSC

nanocircuits with diverse elemental compositions and morphological structures at the nanometer level. Moreover, this work, for the first time, has demonstrated a principal device of NbSe₂ superconductor nanowires and achieved an ultrahigh switch current/hysteresis current ratio, indicating promise for application in superconductor nanowire single-photon detectors (SNSPDs). We believe that the general approach and device demonstration of TMDSC nanocircuits will further promote the practical application of TMDSCs in integrated superconductor electronics, which is the significance of our scientific research.

Comment 3-1: The author demonstrated the process of top-down patterning (TP) and topotactic fabrication (TF) methods in Figure 1. The author claims the patterning process will induce oxidized species and vacancies in the sample. However, the chemical treatments during the patterning process can also induce oxidation of the Nb metal, forming amorphous NbO_x before the chalcogenization process. This pre-contamination of the Nb precursor could further impact the quality of the final NbSe₂ materials. The author needs to clarify the impact of patterning on the Nb precursor and the final NbSe₂ products.

Response: We appreciate the reviewer's insightful comments. To clarify the impact of patterning on the Nb precursor, we have analyzed the compositions of the prepatterned Nb sample by XPS and TOF-SIMS. As shown in **Figure R7a**, both the sputtered Nb film and prepatterned Nb sample suffer from surface oxidation while the latter shows a heavier level due to the patterning treatment. The different fluences of reaching maxima in the TOF-SIMS depth profile (**Figure R7b**) indicate a layered structure of the sample, which is composed of Nb₂O₅, Nb and Al₂O₃ (substrate), supporting surface oxidation in the prepatterned Nb sample. For an accurate description, we have modified the definition of prepatterned metal as a partially-oxidized Nb nanopattern in the revised manuscript.

Figure R7 (Supplementary Fig. S1c,d). (a) Core-level Nb 3d XPS spectra of the Nb film before and after patterning. (b) Normalized TOF-SIMS depth profile of the ultrathin Nb film after nanopatterning.

To clarify the impact of the partially-oxidized Nb nanopattern on the quality of the final NbSe₂ products, XRD and XPS are performed on TF-NbSe₂ and NbSe₂ film. As shown in **Figure R8**, TF-NbSe₂ shows competitive quality in structure and composition to NbSe₂ film. It can be concluded that the partially-oxidized Nb nanopattern can be well and thoroughly transformed into a high-quality NbSe₂ nanopattern through topotactic selenization.

Figure R8 (Supplementary Fig. S8). (a) XRD patterns, and (b) Nb 3d and (c) Se 3d XPS spectra of TF-NbSe₂ and NbSe₂ film.

We have added more discussion on metal degradation and its impact on NbSe₂ quality in the revised manuscript: “The X-ray photoemission spectroscopy (XPS) and time-of-flight secondary ion mass spectrometry results (**Supplementary Fig. S1c,d**) confirm that Nb was partially oxidized into a composite of Nb and Nb₂O₅ after patterning by electron beam lithography and reaction ion etching. Consequently, a partially-oxidized Nb meandered nanowire with a width of 200±10 nm was prepared (**Supplementary Fig. S1e,f**).” (Line 4, Page 5); “Notably, TF-NbSe₂ shows competitive quality in structure and composition to NbSe₂ film prepared by selenization of Nb film (**Supplementary Fig. S8**), indicating that partial oxidation of Nb nanopattern hardly affects the TF-NbSe₂ quality.” (Line 23, Page 7).

Comment 3-2: In Figure 2b-c, the author demonstrated HRTEM images of multilayer TF-NbSe₂. How homogeneous are these results? Does the author also find any region with oxidation? How about the edges of the nanowires (most likely to be influenced by the patterning process)?

Response: We thank the reviewer’s instructive comments. It is undoubted that the homogeneity of the nanowire is critical to the device performance. We have further investigated the homogeneity of TF-NbSe₂ nanowire by Aberration-corrected scanning transmission electron microscopy (STEM). As shown in **Figure R9a**, the cross-sectional image of meandered NbSe₂ nanowire shows a uniform width of 200 nm and the enlarged image shows the continuity and homogeneity of the TF-NbSe₂. The high-resolution STEM image of TF-NbSe₂ (**Figure R9b**) shows the layered structure without any oxidized species, in which five different regions are magnified and present typical atomic arrangement of the NbSe₂ crystal. Even at the edge of TF-NbSe₂ nanowire, as shown in **Figure R9c**, the layered structure is clearly observed without distinguishable oxidation. The corresponding EDS mappings further confirm the intact NbSe₂ composition at the edge of TF-NbSe₂ nanowire with a negligible oxygen signal.

Figure R9. (a) Cross-sectional STEM image of TF-NbSe₂ nanowire. (b) High-resolution STEM image of TF-NbSe₂ and atomic-resolution displays of five regions. (c) STEM image of NbSe₂ nanowire at the edge and corresponding EDS elemental mappings.

We have reorganized **Figure 2** and added **Supplementary Fig. S5** as well as more discussion on the homogeneity of TF-NbSe₂ in the revised manuscript: “Aberration-corrected scanning transmission electron microscopy (STEM) was used to reveal the microstructure of NbSe₂. As shown in **Fig. 2b**, the cross-sectional image of the meandered NbSe₂ nanowire shows a uniform width of 200 nm, and the enlarged image shows the continuity and homogeneity of the TF-NbSe₂. The in-plane HRTEM image (**Fig. 2c**) reveals the in-plane atomic arrangement of the NbSe₂ nanocircuit. The lattice distances were measured to be ~ 0.30 nm, corresponding to the (100) plane. The high-resolution STEM image of TF-NbSe₂ (**Fig. 2d**) shows the layered structure and typical atomic arrangement of the NbSe₂ crystal without any oxidized species. The layer-by-layer van der Waals structure of NbSe₂ with a layer thickness of 0.65 nm and a total thickness of 4.9 nm^{25,41}, is in agreement with the AFM result. Even at the edge of TF-NbSe₂ nanowire, the layered structure is clearly observed without distinguishable oxidation (**Supplementary Fig. S5**). The corresponding EDS mappings further confirm the intact NbSe₂ composition at the edge of TF-NbSe₂ nanowire with a negligible oxygen signal.”. (Line 22, Page 5)

Figure R10 (Fig. 2 in Manuscript) Characterization of TF-NbSe₂ meandered nanowire. (a) SEM image of TF-NbSe₂, in which a gray meandered nanowire on a white background is highlighted by a dashed red circle. (b) Cross-sectional STEM image of TF-NbSe₂ nanowire. (c) In-plane HRTEM image of TF-NbSe₂. (d) High-resolution STEM image of TF-NbSe₂ along with enlarged atomic-resolution display, showing the van der Waals layered structure with a layer distance of 0.65 nm. Core-level Nb 3d (e) and Se 3d (f) XPS spectra of TF-NbSe₂ and NbSe₂ obtained by the top-down patterning method (TP-NbSe₂). (g) Temperature dependence of the resistance for TF-NbSe₂ and TP-NbSe₂.

Figure R11 (Supplementary Fig. S5) Atomic-resolution STEM image of NbSe₂ nanowire at the edge and corresponding EDS elemental mappings.

Comment 3-3: In Figure 2d-e, the author compared the XPS results of TF-NbSe₂ and TP-NbSe₂ and noticed substantial oxidization in the TP-NbSe₂. Can the author also perform a concentration analysis and compare the Nb, Se and O ratios?

Response: We thank the reviewer’s helpful comments. It is noted that the as-grown NbSe₂ samples possess a thickness of approximately 5 nm, which is below the detection depth limitation of XPS; thus, we can rationally analyze the composition based on the Nb 3d XPS spectra. As shown in **Figure R12**, the peaks of NbSe₂ and Nb₂O₅ in the Nb 3d XPS spectra of TF-NbSe₂ and TP-NbSe₂ are deconvoluted. As listed in **Table R2**, TF-NbSe₂ is composed of 93.67% NbSe₂ and 6.33% Nb₂O₅, while TP-NbSe₂ contains 15.4% NbSe₂ and 84.6% Nb₂O₅. The Nb/Se/O ratio is calculated to be 1/1.76/0.30 and 1/0.31/2.12 for TF-NbSe₂ and TP-NbSe₂, respectively.

We have added more discussion on the sample composition in the revised manuscript: “Based on the area of the signal peak after deconvolution (**Supplementary Fig. S7**), TF-NbSe₂ is determined to be composed of 93.67% NbSe₂ and 6.33% Nb₂O₅, while TP-NbSe₂ contains 15.4% NbSe₂ and 84.6% Nb₂O₅. The Nb/Se/O ratios are calculated to be 1/1.76/0.30 and 1/0.31/2.12 for TF-NbSe₂ and TP-NbSe₂, respectively (**Supplementary Table S1**).” (Line 11, Page 7)

Figure R12 (Supplementary Fig. S7). Nb 3d XPS spectra of (a) TF-NbSe₂ and (b) TP-NbSe₂.

Table R2 (Supplementary Table S1). The composition analyses of TF-NbSe₂ and TP-NbSe₂ based on XPS results.

Sample	Area of Nb 3d peaks		Composite ratio	Element content (at%)
	NbSe ₂	Nb ₂ O ₅	NbSe ₂ /Nb ₂ O ₅	Nb/Se/O
TF-NbSe ₂	323,742.1 (88.1%)	43,767.8 (11.9%)	93.67%/6.33%	1/1.76/0.30
TP-NbSe ₂	51,501.4 (15.4%)	281,907.7 (84.6%)	26.69%/73.31%	1/0.31/2.12

Comment 3-4: In Figure 2f, the author demonstrated superconducting behaviors of TF-NbSe₂ and non-superconducting behaviors of TP-NbSe₂. Since the patterning process are almost inevitable during the circuit fabrication process, how do these TP-NbSe₂ results compare to other reports about superconductor nanocircuits patterned from films? Is it possible that other film growth methods (CVD, MOCVD) produce better-quality NbSe₂ materials and, therefore, still achieve good superconductivity behaviour after patterning?

Response: We appreciate the reviewer's insightful comment. As the growth of NbSe₂ film is a challenge and limited (*Nat. Mater.* 2019, 18, 602), to the best of our knowledge, there are no reported works on NbSe₂ nanocircuits patterned from films. Previously, Mills et al. studied the superconducting properties of NbSe₂ nanowires patterned from as-exfoliated NbSe₂ flakes (*Appl. Phys. Lett.* 2014, 104, 052604). Due to the high quality of the single-crystal NbSe₂ flakes, the as-etched TP-NbSe₂ nanowire retained 68.6% of the T_c value of the NbSe₂ flake.

We agree with the reviewer that the better-quality NbSe₂ materials could achieve acceptable superconductivity after top-down patterning. Considering the limited size of single-crystal NbSe₂ exfoliated from the bulk, we have been working on the growth of a single-crystal NbSe₂ wafer to improve the TP-NbSe₂ quality. On the other hand, the quality of TF-NbSe₂ can also be improved by optimizing the fabrication parameters. For instance, using NbN film as the starting material can greatly alleviate the oxidation of the prepatterned precursor due to its better chemical stability than Nb. The XPS results (**Figure R13** and **Table R3**) confirm that the content of Nb₂O₅ in NbN-derived TF-NbSe₂ is much lower than that in Nb-derived TF-NbSe₂.

Figure R13. Nb 3d XPS spectra of (a) TF-NbSe₂ prepared from (a) Nb and (b) NbN.

Table R3. The composition analyses of Nb-TF-NbSe₂ and NbN-TF-NbSe₂ based on XPS results.

Sample	Area of Nb 3d peaks		Composite ratio	Element content (at%)
	NbSe ₂	Nb ₂ O ₅	NbSe ₂ /Nb ₂ O ₅	Nb/Se/O
Nb-TF-NbSe ₂	323,742.1 (88.1%)	43,767.8 (11.9%)	93.67%/6.33%	1/1.76/0.30
NbN-TF-NbSe ₂	302,591.0 (92.7%)	23721.3 (7.3%)	96.21%/3.79%	1/1.85/0.18

Comment 3-5: In most superconductivity measurements, voltage is plotted against current (such as Science, 372(6540), pp.409-412.), and at a certain current, we see a sharp increase in voltage. Higher than that, the circuit shows as a resistor. In this manuscript, Figure 3b plots the current against voltage and demonstrates a sharp increase in current at a certain voltage. Is there a particular reason for that?

Response: Thanks for the reviewer's comments. Compared with the low-resistance superconducting Josephson junction, the normal-state resistance of the nanowire material in the SNSPD is as high as several megaohms, which is larger than the bias

resistor (100 k Ω) in the measurement circuit. Once the bias current exceeds the switch current, the nanowire transforms into the normal state and generates a huge resistance; as a result, the branch current decreases due to the use of a constant voltage source. Therefore, the jump line on the IV curve is due to the sudden change in the resistance of the nanowire. (*Nat. Commun.*, 2022, 13, 5429; *Nat. Nanotechnol.*, 2023, 18, 343).

Figure R14 displays the I-V curves of the 200-nm-wide NbSe₂ device at 300 mK with a current sweep rate of 0.2 $\mu\text{A s}^{-1}$. Clearly, the device remains in the superconducting state until the bias current (I_b) exceeds the switch current (I_{sw} , where the superconducting device transitions to a normal state). Once $I_b > I_{sw}$, the device immediately transforms to a normal state and generates resistance. With decreasing I_b below I_{sw} , the device begins with a normal state due to residual Joule heating and finally recovers to the superconducting state at the hysteresis current (I_h). We have added the scan direction in **Fig. 3b** in the revised manuscript.

Figure R14. I-V curves of 200-nm-wide NbSe₂ device under 300 mK.

REVIEWERS' COMMENTS

Reviewer #1 (Remarks to the Author):

The authors have addressed my questions. It can be published now.

Reviewer #2 (Remarks to the Author):

I have carefully checked the revised version of the paper by Wang et al. The authors have carefully answered those proposed questions raised by the reviewers and revised the manuscript with many necessary data and comments included. I would like to recommend its publication at the current version.

Reviewer #3 (Remarks to the Author):

The authors' responses are very comprehensive and well-organized. It includes all the additional data and detailed explanations I expected to see. I am glad that the oxidation level of the Nb film, patterned Nb, and resulting NbSe₂ pattern is thoroughly discussed, as well as the comparison between TF-NbSe₂ and TP-NbSe₂. In addition, the cross-sectional STEM images are great additions to the manuscript as they demonstrated nice lattices and remarkable homogeneity of the TF-NbSe₂ crystal.

Overall, the rebuttal is an excellent work, significantly improving and enriching the manuscript. Considering the novelty and comprehensiveness of this work, I have no further questions and suggest accepting the manuscript for publication.